evolution, genomics, plant science

plant terrestrialization, plant evolution, streptophytes, green algae, phylogenomics, phycology

**Authors for correspondence:**
Iker Irisarri
e-mail: irisarri.iker@gmail.com
Tatyana Darienko
e-mail: tdarien@gwdg.de
Jan de Vries
e-mail: devries.jan@uni-goettingen.de

†These authors contributed equally to this study.

# Unexpected cryptic species among streptophyte algae most distant to land plants

Iker Irisarri[1,2,†], Tatyana Darienko[1,5,†], Thomas Pröschold[4],
Janine M. R. Fürst-Jansen[1], Mahwash Jamy[6] and Jan de Vries[1,2,3]

[1]Department of Applied Bioinformatics, Institute for Microbiology and Genetics, [2]Campus Institute Data Science (CIDAS), and [3]Göttingen Center for Molecular Biosciences (GZMB), Department of Applied Bioinformatics, University of Goettingen, Goldschmidtstrasse 1, 37077 Göttingen, Germany
[4]Research Department for Limnology, Leopold-Franzens-University of Innsbruck, Mondseestrasse 9, 5310 Mondsee, Austria
[5]Albrecht-von-Haller-Institute of Plant Sciences, Experimental Phycology and Culture Collection of Algae, University of Goettingen, Nikolausberger Weg 18, 37073 Göttingen, Germany
[6]Department of Organismal Biology, Program in Systematic Biology, Uppsala University, Norbyvägen 18D, 75236 Uppsala, Sweden

II, 0000-0002-3628-1137; TD, 0000-0002-1957-0076; TP, 0000-0002-7858-0434;
JMRF-J, 0000-0002-5269-8725; MJ, 0000-0002-2930-9226; JdV, 0000-0003-3507-5195

Streptophytes are one of the major groups of the green lineage (Chloroplastida or Viridiplantae). During one billion years of evolution, streptophytes have radiated into an astounding diversity of uni- and multicellular green algae as well as land plants. Most divergent from land plants is a clade formed by Mesostigmatophyceae, *Spirotaenia* spp. and Chlorokybophyceae. All three lineages are species-poor and the Chlorokybophyceae consist of a single described species, *Chlorokybus atmophyticus.* In this study, we used phylogenomic analyses to shed light into the diversity within *Chlorokybus* using a sampling of isolates across its known distribution. We uncovered a consistent deep genetic structure within the *Chlorokybus* isolates, which prompted us to formally extend the Chlorokybophyceae by describing four new species. Gene expression differences among *Chlorokybus* species suggest certain constitutive variability that might influence their response to environmental factors. Failure to account for this diversity can hamper comparative genomic studies aiming to understand the evolution of stress response across streptophytes. Our data highlight that future studies on the evolution of plant form and function can tap into an unknown diversity at key deep branches of the streptophytes.

## 1. Background

Green algae and land plants (Chloroplastida or Viridiplantae) consist of three major lineages: the recently pinpointed Prasinodermophyta [1], Chlorophyta and Streptophyta [2]. Streptophyta are about a billion years old [3,4] and encompass the main constituents of the land flora, the Embryophyta (land plants). In addition, Streptophyta include the algal relatives of land plants, known as streptophyte algae. In the past few years, the phylogenetic backbone of the green lineage has been brushed up. This was largely thanks to both an increased effort in sequencing streptophyte algae [5–13] and the use of these data in phylogenomic analyses to infer a robust green tree of life [2,14,15]. The new phylogenetic framework marked a milestone; it clarified the phylogenetic relationships among land plants and their streptophyte algal relatives. Within streptophytes, the position of Zygnematophyceae as closest relatives to land plants made quite a splash. However, equally important was the

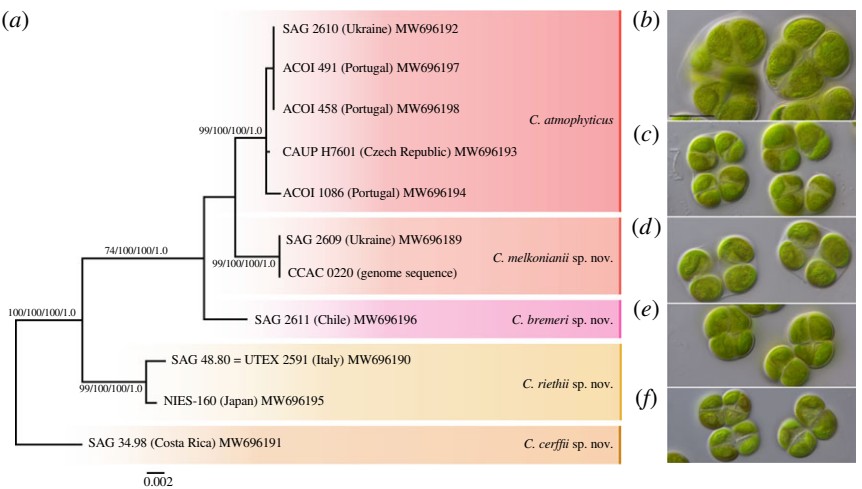

**Figure 1.** Cryptic diversity in *Chlorokybus*. (*a*) Maximum-likelihood phylogeny of SSU + ITS rDNA from all eleven isolates currently available in culture (root *sensu* figure 3). Branch support are respectively non-parametric bootstrap proportions from neighbour-joining, maximum parsimony, maximum likelihood and Bayesian posterior probabilities and branch lengths are in expected substitutions per site. Light micrographs correspond to: (*b*) *C. atmophyticus* ACOI 1086, (*c*) *C. melkonianii* sp. nov. SAG 2609, (*d*) *C. bremeri* sp. nov. SAG 2611, (*e*) *C. riethii* sp. nov. NIES-160, (*f*) *C. cerffii* SAG 34.98. Scale bar = 10 µm. (Online version in colour.)

recovery of Mesostigmatophyceae, *Spirotaenia* spp. [2] and Chlorokybophyceae as sister to all other Streptophyta [16]. Both Chlorokybophyceae and Mesostigmatophyceae are thought to encompass, respectively, one or few extant species. The apparent low diversity in these key lineages complicates macroevolutionary studies that aim to reconstruct the early evolution of key traits in the streptophyte ancestor. Recent genomic and phylogenomic investigations have honed in on freshwater and terrestrial streptophyte algae because they provide important insights into the origin of land plants and the evolution of response mechanisms to terrestrial stressors [2,5,7,10,12,13,17].

Here, we investigate the diversity within the Chlorokybophyceae using a phylotranscriptomic approach with broad sampling of isolates across its known distribution (Eurasia, Central and South America). We pinpoint that the Chlorokybophyceae consist of a cryptic species complex of at least five extant members.

## 2. Results and discussion

### (a) Chlorokybophyceae is an oligotypic class

Chlorokybophyceae is thought to be a monotypic class with a single described species, *Chlorokybus atmophyticus* Geitler 1942. *Chlorokybus* is a subaerial alga inhabiting soil and rock surfaces and cracks [18–21]; it has been isolated from Europe and Central and South America, although it is thought to have a cosmopolitan distribution, despite being rare (electronic supplementary material, 'Portrait and history of *Chlorokybus*'). To further explore the distribution and diversity of *Chlorokybus*, we searched four large soil environmental sequencing datasets (Neotropical forest, Swiss Alps, meadow and agricultural soils from the UK and Tibet, and a set of globally distributed soils; approximately 128 Mio. reads total). Only a single amplicon sequence variant (ASV) of *Chlorokybus* was obtained, which was composed of 32 reads total (less than 0.01% abundance; electronic supplementary material, table S1). This ASV originated from a high-altitude Swiss Alpine soil sample [22]. Phylogenetic analyses confirmed the identity of this ASV as *Chlorokybus*, but its precise phylogenetic position could not

be determined because the SSU V4 region has limited phylogenetic signal [23] (electronic supplementary material, figure S1). None of the primer sets used in the above studies were biased against *Chlorokybus* and DNA extraction methods are unlikely to be so, but the lack of rocky outcrop samples in the above studies could have exacerbated the reported low abundance. Currently, 11 strains of *Chlorokybus* are available in public culture collections, none of them were isolated from the type locality and therefore no authentic strain is available (electronic supplementary material, table S2). We performed a phylogenetic analysis including all available *Chlorokybus* strains with two commonly used nuclear markers (SSU and ITS rDNA). This phylogeny suggested a deep genetic structure within *Chlorokybus* (figure 1*a*). Extensive observations under light microscope revealed no obvious morphological differences among the studied isolates, despite marked genetic divergences: all studied *Chlorokybus* isolates form sarcinoid, cubical packets of two to eight cells with a gelatinous matrix; cells are spherical or broadly ellipsoidal and contain a parietal slightly lobated chloroplast with two types of pyrenoids (figures 1 and 2; electronic supplementary material, figure S3; full description is provided below). The life cycle is haploid and was studied by Rieth [21] (figure 2). Since the phenotype did not give away hints as to the differences among the *Chlorokybus* strains, we garnered more sequence data.

### (b) A phylotranscriptomic framework for *Chlorokybus*

Using the Illumina NovaSeq6000 platform, we generated 224 million paired-end reads (greater than 47 Gbp of raw sequence information) on four isolates of *Chlorokybus* from across its known distribution range. Combining these data with published genomic and transcriptomic information from other algae and land plants (electronic supplementary material, table S3), we inferred a robust phylogenomic tree based on 529 densely sampled loci (17% missing data). The maximum-likelihood tree, which was inferred with IQ-TREE under the LG + F + I + $\Gamma$4 + C60 mixture model, unambiguously recapitulated the accepted phylogeny of the green lineage (Chloroplastida), including the position of *Chlorokybus* (Chlorokybophyceae), *Mesostigma* (Mesostigmatophyceae) and *Spirotaenia minuta* as the sister group to all other

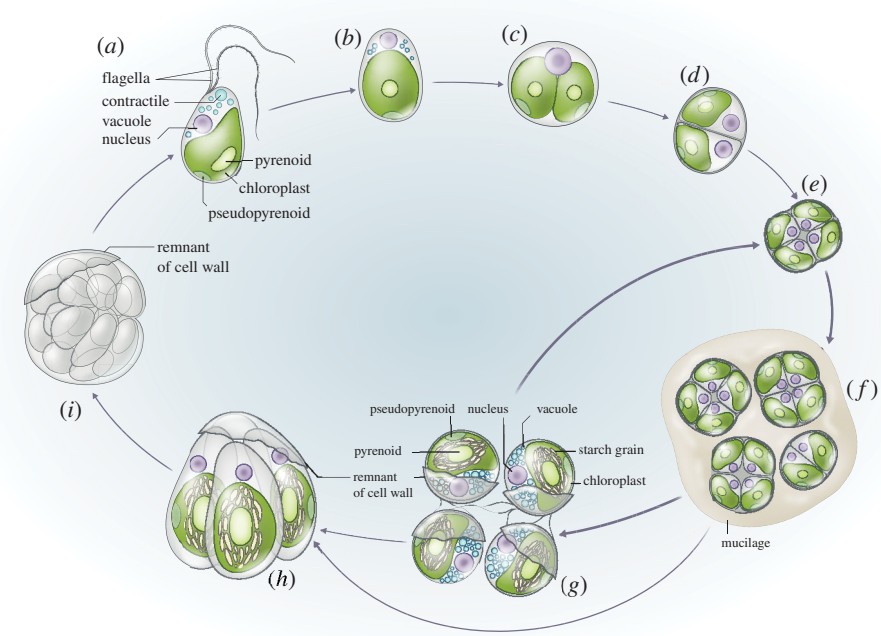

**Figure 2.** Life cycle of *Chlorokybus*. (*a*) Zoospore with two unilaterally inserted flagella in slightly under apical position. (*b*) A young vegetative cell is formed after the zoospore is settled and (*c*) cell division can begin. (*d*) Two-cell stage of daughter cells are contained within the same gelatinous matrix and (*e*) cubic cell packages can contain groups of two to eight cells each. (*f*) Mature packages produce mucilage and (*g*) cell cycle can proceed through the production of autospores for asexual reproduction (*g* to *e*). (*h*) Zoospores might be formed by differentiation from autospores (*g* to *h*) or directly from mature packages (*f* to *h*). (*i*) Zoospores can form groups of up to 32 cells called 'Maulbeerstadium'. Cell cycle based on Rieth [21]. (Online version in colour.)

streptophytes [1,13,14] (figure 3*a*). A summary coalescent analysis recovered an almost identical tree topology, except the interrelationships between SAG 34.98 (*C. cerffii* sp. nov.) and NIES-160 and UTEX 2591 (*C. riethii* sp. nov.) could not be resolved with certainty (electronic supplementary material, figure S2). Our phylotranscriptomic trees show unmistakable deep genetic structure within *Chlorokybus*, represented here by eight isolates. The genetic distances among *Chlorokybus* isolates are often more than twice as those recovered among three different species of *Arabidopsis* (figure 3*a*). The inferred patristic (maximum-likelihood) distances among *Chlorokybus* species are between 0.0254 and 0.0874 substitutions per site (*p*-uncorrected distances: 0.0245–0.0730), whereas the distances among the three *Arabidopsis* species are between 0.0149 and 0.0346 (*p*-uncorrected distances: 0.0147–0.0332) (table 1). A Bayesian relaxed molecular clock analysis calibrated with eight fossils (uniform priors) found that divergences within *Chlorokybus* could be as old as 76 Ma (95% HPD interval: 54–102 Ma) and the divergence between the two closest isolates described here as species—*C. atmophyticus* and *C. melkonianii* sp. nov., see below—was 24 Ma (95% HPD 15–34 Ma) (electronic supplementary material, figure S4). The use of more informative prior distributions for fossil calibrations (*t*-cauchy and skew-*t*) produced slightly younger divergences, as expected, but differences were not substantial (average differences within *Chlorokybus* were 0.47 and 1.47 Ma, respectively) (electronic supplementary material, figure S4). In contrast to *Chlorokybus*, the divergences among *Arabidopsis* species were 13 Ma (95% HPD 7–19 Ma) and 28 Ma (95% HPD 18–39 Ma).

To further scrutinize the deep genetic structure within *Chlorokybus*, we performed a maximum-likelihood analysis of 75 plastid proteins using IQ-TREE and the best-fit cpREV + F + I + $\Gamma$4 + C60 mixture model. The plastid phylogeny was moderately resolved and statistically supported; it further confirmed the deep divergences among *Chlorokybus* isolates, even though internal relationships in *Chlorokybus* differed from the nuclear tree (electronic supplementary material, figure S5). Similar plastid-nuclear incongruences are often observed in algae, for example in Volvocales [24], and might be due to either methodological or biological reasons (e.g. introgression), or both. While biological confounding factors cannot be excluded, the failure to recover *Amborella* as sister to all other flowering plants suggests the presence of biases and/or limited phylogenetic signal in the plastid dataset. At any rate, both plastid and nuclear marker phylogenies agreed on the presence of deep divergences among *Chlorokybus* isolates.

The final assessment of the genetic diversity within *Chlorokybus* is based on the more robust nuclear phylotranscriptomic dataset. On the basis of the inferred deep divergences, we propose a new taxonomic arrangement by describing four new species and assigning a lectotype and an epitype for *C. atmophyticus*, for which no authentic strain is available in public culture collections (see 'Systematic botany').

Taking advantage of the fact that the new isolates were grown simultaneously under the same experimental conditions, we explored whether the genetic distances among species are reflected in differences in global gene expression patterns. Clean Illumina reads were mapped against the annotated *Chlorokybus* genome using STAR [25] and expression quantified with RSEM [26], followed by TMM (trimmed mean of *M*-values) cross-sample normalization. While the lack of biological replicates prevented us from inferring differential gene expression, we observed marked differences in steady-state gene expression levels among the four new isolates (figure 3*b*, *c*). The clustering of expression values mirrored the species phylogeny, with NIES-160 (*C. riethii* sp. nov.) showing the most

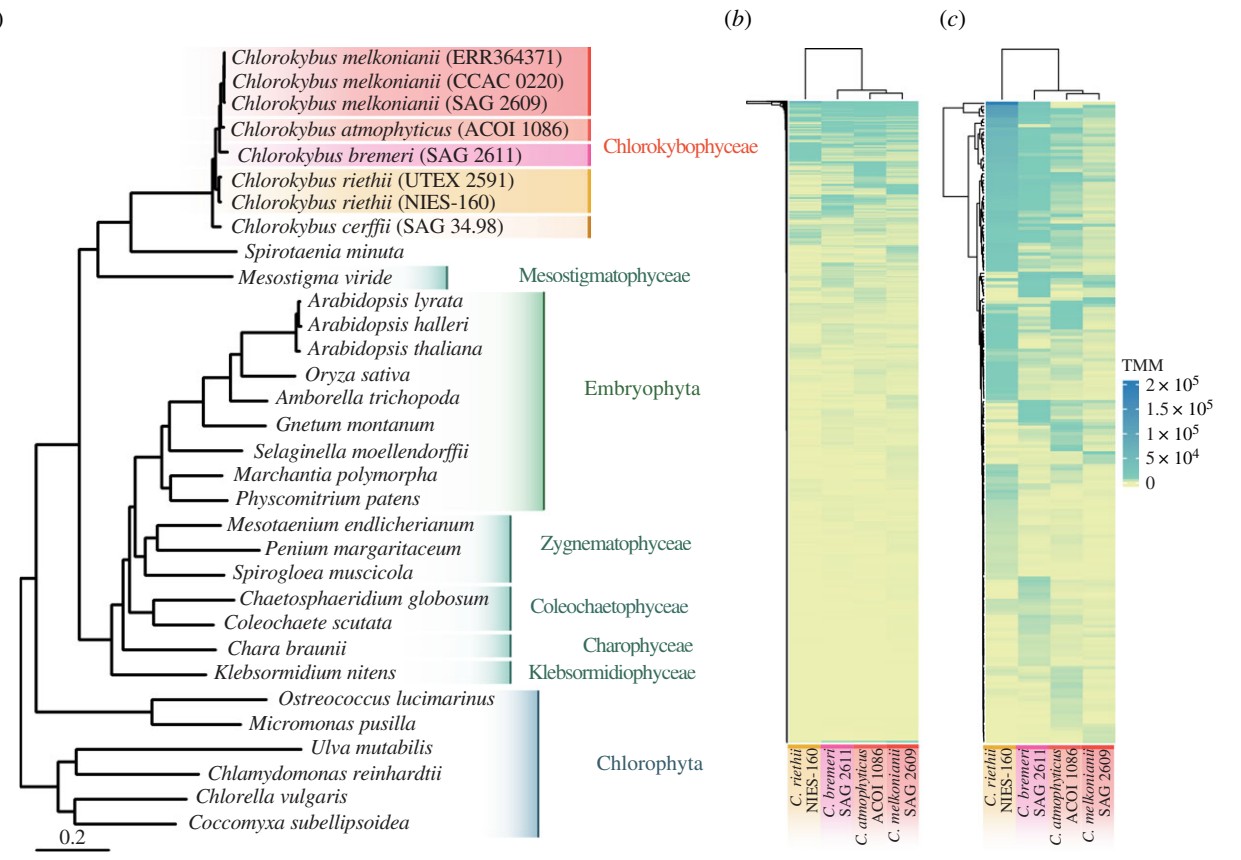

**Figure 3.** Transcriptomic evidence for deep phylogenetic divergences and expression differences within *Chlorokybus*. (*a*) Maximum-likelihood phylogeny based on 529 densely sampled loci, inferred with IQ-TREE under $LG + F + I + \Gamma 4 + C60$ and support values from 1000 pseudoreplicates of UFBoot2 and SH-aLRT (all branches received 100% support). Branch lengths are in expected substitutions per site. (*b,c*) Gene expression differences (TMM, trimmed mean of *M*-values) among four isolates grown simultaneously under the same experimental conditions. Heatmaps correspond to (*b*) the 9300 annotated proteins in the *C. melkonianii* genome (no filtering) and (*c*) the top 200 proteins with the highest expression differences. (Online version in colour.)

**Table 1.** Genetic distances among *Chlorokybus* isolates and *Arabidopsis* species measured from concatenated amino acid alignments of 529 loci (178 397 aligned amino acids). *p*-uncorrected (upper triangle) and maximum-likelihood distances (lower triangle; figure 1) are shown, with intra-specific comparisons in italics.

|  |  | 1 | 2 | 3 | 4 | 5 | 6 | 7 | 8 |
|---|---|---|---|---|---|---|---|---|---|
| 1 | C. cerffii (SAG 34.98) | | 0.0573 | 0.0619 | 0.0781 | 0.0690 | 0.0718 | 0.0730 | 0.0722 |
| 2 | C. riethii (NIES-160) | 0.0621 | | *0.0133* | 0.0646 | 0.0507 | 0.0482 | 0.0539 | 0.0522 |
| 3 | C. riethii (UTEX 2591) | 0.0677 | *0.0135* | | 0.0648 | 0.0501 | 0.0501 | 0.0529 | 0.0517 |
| 4 | C. bremeri (SAG 2611) | 0.0874 | 0.0710 | 0.0713 | | 0.0424 | 0.0446 | 0.0497 | 0.0486 |
| 5 | C. atmophyticus (ACOI 1086) | 0.0762 | 0.0543 | 0.0536 | 0.0452 | | 0.0245 | 0.0289 | 0.0277 |
| 6 | C. melkonianii (ERR364371) | 0.0798 | 0.0514 | 0.0536 | 0.0479 | 0.0254 | | *0.0044* | *0.0002* |
| 7 | C. melkonianii (SAG 2609) | 0.0811 | 0.0580 | 0.0569 | 0.0537 | 0.0302 | *0.0045* | | *0.0037* |
| 8 | C. melkonianii (CCAC 0220) | 0.0801 | 0.0559 | 0.0553 | 0.0523 | 0.0288 | *0.0002* | *0.0037* | |

|  |  | 1 | 2 | 3 |
|---|---|---|---|---|
| 1 | A. halleri | | 0.0332 | 0.0147 |
| 2 | A. thaliana | 0.0346 | | 0.0288 |
| 3 | A. lyrata | 0.0149 | 0.0296 | |

different expression profile, followed by SAG 2611 (*C. bremeri* sp. nov.), and the more similar profiles shown by ACOI 1086 (*C. atmophyticus*) and SAG 2609 (*C. melkonianii* sp. nov.). Yet, even the two latter isolates showed marked differences in gene expression, which together with the reported genetic distances support the notion that they are not only different species but might also exhibit different cell physiologies.

## 3. Conclusion

Here, we report on the presence of consistent deep structure within *Chlorokybus* after analysing all currently available isolates. These divergences might date back to approximately 76 Ma and are twice as large as those among some flowering plant species (e.g. *Arabidopsis*). Deep genetic divergences

among *Chlorokybus* isolates are further supported by substantial gene expression variation when grown under the same experimental conditions. Yet, these genetic differences are not reflected in appreciable morphological differences, which suggest the presence of undescribed cryptic diversity within this lineage. All this genetic diversity has remained unnoticed under the umbrella name *Chlorokybus atmophyticus*, the only validly described species so far. To remedy this, we describe four new species of *Chlorokybus* and designate a cryopreserved culture as epitype for *C. atmophyticus*. *Chlorokybus* species are probably cosmopolitan but rare, as further supported by our search across global soil metabarcoding datasets that identified a single sequence of this genus. Properly recognizing the existing diversity within *Chlorokybus* is paramount, given the key phylogenetic position of Chlorokybophyceae, which together with *Spirotaenia* spp. [27] and Mesostigmatophyceae are the sister lineage to all other streptophytes. This diversity has to be taken into account for the adequate comparison of current and future data from different *Chlorokybus* strains [2,8,13]. In fact, the reported gene expression differences might even suggest certain interspecific variability in responding to environmental factors and adequately accounting for this will be essential in comparative genomic studies that aim to understand the evolution of key traits (such as phytohormone or stress response pathways [17]) along the backbone phylogeny of streptophytes. Our phylogenetic analysis of genomic data can aid in uncovering key cryptic diversity, which together with the discovery of new deep-branching lineages [28–30], are revealing important pieces in the puzzle that is the Eukaryotic Tree of Life.

## 4. Systematic botany

In the following, we describe four new species of *Chlorokybus* and designate a lectotype and an epitype for *C. atmophyticus*, given that no cultured material is available from the different locations studied by Geitler [18–20]. We further provide a formal description of the class Chlorokybophyceae, which was originally proposed by Bremer [31] without formal description nor page numbers, and thus being invalid under articles 38.1 and 41.5 of the International Code of Nomenclature (ICN) for algae, fungi and plants [32].

**Class Chlorokybophyceae** class. nov. (figure 2)

Description: Green algae forming sarcinoid, cubical packets. Single chloroplast containing two pyrenoids. First pyrenoid located in the middle of the chloroplast and surrounded by starch grains. Second naked pyrenoid (or called pseudopyrenoid) located at the edge of the chloroplast. Reproduction can occur asexually by breaking cell packages into separate cells or by zoospores (figure 2). Zoospores are produced one per cell and possess two laterally inserted flagella. The flagella and body are covered with square scales. The flagellar apparatus is non-cruciate unilateral and contains multi-layered structures (MLS). After settling of the zoospores, the flagella are retracted at the point of their insertion. Cell division type phragmoplast-like, presence of advanced cleavage furrow and VII type of mitosis (*sensu* van den Hoek *et al.* [33]). Sexual reproduction is not observed. The class is supported by SSU rDNA, plastid and nuclear transcriptomic data.

Type order (designated here): Chlorokybales ordo nov.

**Order Chlorokybales** ordo nov.

Description: With features of the class.

Type family (designated here): Chlorokybophyceae fam. nov.

**Family Chlorokybaceae** fam. nov.

Description: With features of the class.

Type genus (designated here): *Chlorokybus* Geitler 1942, Österr. Bot. Z. 91: 51.

Comment: Rogers *et al.* (1980) proposed the order Chlorokybales and the family Chlorokybaceae without Latin diagnosis. They referred to the Latin diagnosis of Geitler (1942/43), but he published the Latin diagnosis in 1942 (see detailed citation below).

Type species: *Chlorokybus atmophyticus* Geitler 1942, Österr. Bot. Z. 91: 51; Geitler 1942/43, Flora 136, fig. 2 (lectotype designated here).

Emended description: Cell size 16.9–20.0 µm length × 12.0–16.5 µm wide. Other features are identical to the class description. SSU-ITS sequence (MW696194) and NCBI BioSample accession SAMN18221336 (RNA-Seq), ITS-2 Barcode: BC-1 in electronic supplementary material, figure S6.

Diagnosis: Differs from other species of *Chlorokybus* by SSU-ITS and transcriptome sequence.

Epitype (designated here): Strain ACOI 1086 cryopreserved in metabolically inactive state at the Culture Collection of Algae (SAG), Georg-August-University Göttingen, Germany (figure 1*c*; electronic supplementary material, figure S3*a–d*).

*Chlorokybus melkonianii* sp. nov.

Description: Cell size 10.3–13.5 µm length × 7.7–10.6 µm wide. Other features are identical to the class description. SSU-ITS sequence (MW696189), NCBI BioSample accession SAMN18221334 (RNAseq), ITS-2 Barcode: BC-2 in electronic supplementary material, figure S6.

Diagnosis: Differs from other species of *Chlorokybus* by SSU-ITS and transcriptome sequence.

Holotype (designated here): Strain SAG 2609 cryopreserved in metabolically inactive state at the Culture Collection of Algae (SAG), Georg-August-University Göttingen, Germany (figure 1*b*; electronic supplementary material, figure S3*e–h*).

Type locality: Europe, Ukraine, regional landscape park 'Trakhtemyriv', sandstone outcrops, in crack.

Etymology: The species epithet honours Prof. Dr Michael Melkonian (University of Cologne, Germany) for his important contributions to understanding the diversity and evolution of algae.

Comment: The strain CCAC 0220 represents another isolate of this species and the SSU-ITS sequence and NCBI BioSample accession are available under SAMEA2242428 (RNAseq) and SAMN10351691 (genome assembly), respectively.

*Chlorokybus bremeri* sp. nov.

Description: Cell size 13.1–16.8 µm length × 9.7–11.5 µm wide. Other features are identical to the class description. SSU-ITS sequence (MW696196) and NCBI BioSample accession SAMN18221335 (RNA-Seq), ITS-2 Barcode: BC-3 in electronic supplementary material, figure S6.

Diagnosis: Differs from other species of *Chlorokybus* by SSU-ITS and transcriptome sequence.

Holotype (designated here): Strain SAG 2611 cryopreserved in metabolically inactive state at the Culture Collection of Algae (SAG), Georg-August-University Göttingen, Germany (figure 1*d*; electronic supplementary material, figure S3*i–l*).

Type locality: South America, Chile, national park 'La Campana', granite outcrops, in crack.

Etymology: The species epithet honours Prof. Dr Kåre Bremer (University of Stockholm, Sweden), who first proposed the class name Chlorokybophyceae.

*Chlorokybus riethii* sp. nov.

Description: Cell size 13.1–16.8 µm length × 9.7–11.5 µm wide. Other features are identical to the class description. SSU-ITS sequence (MW696190) and NCBI accession SRX025846 (RNA-Seq), ITS-2 Barcode: BC-4a/b in electronic supplementary material, figure S6.

Diagnosis: Differs from other species of *Chlorokybus* by SSU-ITS and transcriptome sequence.

Holotype (designated here): Strain SAG 48.80 cryopreserved in metabolically inactive state at the Culture Collection of Algae (SAG), Georg-August-University Göttingen, Germany (figure 1*e*; electronic supplementary material, figure S3*m–r*).

Type locality: Europe, Italy, Neaples, in enrichment culture of *Porphyridium purpureum* from the Mediterranean Sea.

Etymology: The species epithet honours Prof. Dr Alfred Rieth (Institute for Genetics and Crop Plant Research Gatersleben) for his detailed observations of *Chlorokybus*.

Comment: The strain NIES-160 represents another isolate of this species and the SSU-ITS sequence and NCBI Bio-Sample accession are available under MW696195 and SAMN18221337 (RNA-Seq), respectively.

*Chlorokybus cerffii* sp. nov.

Description: Cell size 13.1–16.8 µm length × 9.7–11.5 µm wide. Other features are identical to the class description. SSU-ITS sequence (MW696191) and NCBI BioSample accession SAMN07525888 (RNA-Seq), ITS-2 Barcode: BC-5 in electronic supplementary material, figure S6.

Diagnosis: Differs from other species of *Chlorokybus* by SSU-ITS and transcriptome sequence.

Holotype (designated here): Strain SAG 34.98 cryopreserved in metabolically inactive state at the Culture Collection of Algae (SAG), Georg-August-University Göttingen, Germany (figure 1*f* and electronic supplementary material, figure S3*s–v*).

Type locality: Central America, Costa Rica, Province Heredia, Barreal coffee plantation, soil.

Etymology: The species epithet honours Prof. Dr Rüdiger Cerff (Braunschweig University of Technology, Germany) for his contributions on endosymbiosis research and plant evolutionary biology.

# 5. Material and methods

## (a) Culturing conditions

Details about isolate origins are available in electronic supplementary material, table S2. Four isolates (NIES-160, SAG 2611, ACOI 1086, SAG 2609) were cultivated on 3N-BBM + V medium (medium 26a in Schlösser [34]) at 18°C, with 20 µmol photons $m^{-2} s^{-1}$ provided by daylight fluorescent tubes (TL-D 18 W 640, Osram, Munich, Germany), and light : dark cycle of 16 : 8 h. Data for the strain SAG 34.98 were obtained from de Vries *et al.* [8], cultured in ES (medium 1 in Schlösser [35]) at 20°C with 50 µmol of photons $m^{-2} s^{-1}$ from LED light source and light : dark cycle of 12 : 12 h. Three-week-old cultures were used for morphological identification, comparing them to the original species descriptions. Light microscopy used an Olympus BX-60 microscope (Olympus, Tokyo, Japan), a ProgRes C14plus camera, and the ProgRes CapturePro imaging system (v2.9.0.1) (Jenoptik, Jena, Germany).

## (b) rDNA phylogeny

DNA was extracted with the DNeasy Plant Mini kit (Qiagen, Hilden, Germany) following the manufacturer's instructions. The SSU and ITS were amplified using the *Taq* PCR MasterMix Kit (Qiagen) with primers EAF3 and ITS055R [36]. PCR reactions had initial denaturation at 95°C for 5 min followed by 30 cycles of 1 min at 95°C, 2 min at 55°C and 3 min at 68°C, and a final step at 68°C for 10 min. PCR products were purified and sequenced as in Darienko *et al.* [37]. A multiple sequence alignment of SSU was performed according to the predicted secondary structures (electronic supplementary material, figure S6). ITS-1 and ITS-2 were folded according to Darienko *et al.* [38]. SSU/ITS sequences were concatenated into a dataset containing 11 OTUs (2,424 bp). We used PAUP 4.0a build 169 [39] to select the best-fit evolutionary model (GTR + I) according to the Akaike information criterion (AICc). Neighbour-joining, maximum parsimony, maximum likelihood and Bayesian inference were conducted following Darienko *et al.* [34], using PAUP v4.0a build 169 [39], RAxML v8.2.12 [40], MrBayes v3.2.7a using the doublet approach [41] and PHASE package v2.0 [42].

## (c) Metabarcoding

Environmental eukaryotic SSU amplicon sequences were obtained from previous studies [22,43–45] (electronic supplementary material, table S1). Short-read data were cleaned and denoized into Amplicon Sequence Variants (ASV) using DADA2 [46]. SSU of the *C. mekonianii* genome (RHPI01002076.1:1257-3060) was used to search the datasets with BLASTN v2.11.0+ [47] using a 95% sequence similarity threshold. Primers from the original studies were tested *in silico* with the TestPrime function in SILVA [48] to discard biases against *Chlorokybus*. The identified *Chlorokybus* ASV was aligned to other *Chlorokybus* rDNA using MAFFT v7.304b [49] (default settings) and a phylogeny was inferred with IQ-TREE v1.6.12 [50] using the BIC-selected model and 1000 non-parametric bootstrapping pseudoreplicates.

## (d) RNAseq and transcriptome assembly

Algae were scraped off the agar and transferred into 1 ml Trizol (Thermo Fisher, Carlsbad, CA, USA), ground using a Tenbroek tissue homogenizer and RNA isolated according the manufacturer's instructions. RNA samples were treated with DNAse I (Thermo Fisher, Waltham, MA, USA) and quality and quantity assessed with a formamide agarose gel and Nanodrop (Thermo Fisher), respectively. RNA was shipped on dry ice to Genome Québec (Montreal, Canada). After Bioanalyzer (Agilent Technologies Inc., Santa Clara, CA, USA) quality check, libraries were built using the NEBNext mRNA stranded library preparation kit (New England Biolabs, Beverly, MA, USA). Libraries were checked with Bioanalyzer and sequenced using NovaSeq 6000 (Illumina) with NEBNext dual adapters: 5′-AGATCGGAAGAGC ACACGTCTGAACTCCAGTCAC-3′ for read 1 and 5′-AGATCGG AAGAGCGTCGTGTAGGGAAAGAGTGT-3′ for read 2. FastQC (www.bioinformatics.babraham.ac.uk/projects/) reports are available in Dryad.

We downloaded RNAseq data for *Chlorokybus atmophyticus* SAG 34.98 [8] (SRX3107749-SRX3107751), *Chlorokybus melkonianii* [2] (ERR364371), *Chaetosphaeridium globosum* [2] (ERR364369), and *Coleochaete orbicularis* [51] (SRR1594679). For all samples, transcriptomes were assembled *de novo* using Trinity v2.11.0 [52] after adapter trimming (—trimmomatic). SuperTranscripts [53] were inferred by collapsing splicing isoforms, as implemented in Trinity. Transcriptome completeness was assessed with BUSCO v4.1.0 [54] with the 'chlorophyta_odb10' reference set. All new assemblies recovered greater than 75% complete BUSCOs (electronic supplementary material, table S4). Protein-coding genes were identified with Transdecoder v5.5.0 using *Chlorokybus*'s

annotated proteins (CCAC 0220) as reference in BLASTP searches and retaining only the longest open reading frame (ORF) per transcript (—single_best_only). A total of 19 147 transcripts for *C. riethii* UTEX 2591 (published as *C. atmophyticus* [55]) were downloaded from GenBank, assembled from GS FLX Titanium 454.

## (e) Phylotranscriptomic dataset construction

Likely contaminants were removed by sequence similarity searches against a database containing proteins from (i) *Chlorokybus melkonianii* (CCAC 0220) [13] and possible contaminants including (ii) RefSeq representative bacterial genomes (11 318 genomes) and (iii) fungi (2397), (iv) all available viruses (125), (downloaded from GenBank on 17/08/2020), (v) *Chlamydomonas reinhardtii*, and (vi) a *Vermamoeba vermiformis* transcriptome [56]. *Vermamoeba* and *Chlamydomonas* were identified as likely contaminants in some cultures. MMseqs2 [57] was used with an iterative search with increasing sensitivities, real sequence identity, and keeping 10 hits maximum (--start-sens 1 --sens-steps 3 -s 7 --alignment-mode 3 --max-seqs 10). As strict decontamination criterion, only sequences whose best hit corresponded to a predicted *Chlorokybus* nuclear proteins were kept for phylogenetic analyses (4817–19 566 proteins per species; 8690–10 917 for new transcriptomes).

## (f) Phylotranscriptomic phylogeny

A representation of chlorophytes and streptophytes were used as outgroups (electronic supplementary material, table S3). Orthofinder v2.4.0 [58] was used to infer orthogroups using a species tree following Leebens-Mack *et al.* [2] with unresolved relationships within *Chlorokybus*. We selected phylogenetic hierarchical orthogroups at the tree root [58]. In total, 2386 orthogroups contained data for all major lineages (in practice, at least one sequence each for *Chlorokybus*, *Mesostigma* or *Spirotaenia minuta*, *Coleochaete* or *Chaetosphaeridium*, *Chara* or *Klebsormidium*, Zygnematophyceae, chlorophytes, bryophytes, and vascular plants). Homologous sets were aligned with MAFFT v7.304b [49] (default settings) and subjected to maximum-likelihood inference with IQ-TREE v1.6.12 [50] using fast searches, BIC-selected best-fit nuclear models, and SH-like aLRT branch support (-m TEST -msub nuclear -fast -alrt 1000). Phylopyrpruner v1.2.3 (Thalen *et al.*, https://pypi.org/project/phylopypruner/) was used to prune orthologue sets (--prune MI --mask pdist --trim-lb 5 --trim-freq-paralogues 4 --trim-divergent 1.25 --min-pdist 1 × $10^{-8}$ --min-support 0.75 --min-taxa 10 --min-gene-occupancy 0.1 --min-otu-occupancy 0.1), resulting in 946 orthologue sets. After applying the above taxonomic filter, we selected 529 final loci, which were masked with PREQUAL v1.02 [59], aligned with MAFFT ginsi v7.304b with a variable scoring matrix ('--allowshift --unalignlevel 0.8') [60], and columns greater than 75% gaps removed with ClipKIT v0.1 [61]. Trimmed alignments were concatenated into a matrix containing 32 taxa and 529 loci (17% missing sequences) and 178 397 aligned amino acid positions. Maximum-likelihood trees were inferred using IQ-TREE under BIC-selected homogeneous (LG + F + I + $\Gamma$4) and mixture (LG + F + I + $\Gamma$4 + C60) models and branch support assessed with 1000 pseudoreplicates of UFBoot2 [62] and SH-like aLRT [63]. ASTRAL v5.7.5 [64] was run on gene trees inferred by IQ-TREE with BIC-selected models (branches with less than 10% bootstrap were collapsed). P-uncorrected distances were calculated with MEGA X v10.2.4 [65] on the phylotranscriptomic dataset, whereas patristic distances were obtained from the LG + F + I + $\Gamma$4 + C60 tree.

## (g) Relaxed molecular clock

Bayesian molecular dating was performed with MCMCTree [66] within the PAML package v4.9 h [67]. We used the phylotranscriptomic tree (figure 3) and eight fossil calibrations with

uniform, t-cauchy and skew-t prior distributions, following parametrizations in Morris *et al.* [3] (their electronic supplementary material, table S8). We assumed relaxed uncorrelated lognormal molecular clocks (clock = 2) and birth–death tree priors. Analyses used approximate-likelihood calculations [68] on the phylotranscriptomic dataset (single partition) under the LG + $\Gamma$ model. A diffuse gamma Dirichlet prior was used for the prior on mean rates as 0.1407 replacements site$^{-1}$ $10^8$ Myr$^{-1}$ ('rgene_gamma'; $\alpha = 2$, $\beta = 14.21$). The rate drift parameter reflected considerable rate heterogeneity across lineages ('sigma2_gamma'; $\alpha = 2$, $\beta = 2$). A 100 Ma time unit was assumed. Two independent MCMC chains were run for each analysis, consisting of 22 million generations and the first 2 000 000 were excluded as burnin. Convergence was checked using Tracer v1.7.1 [69]; all parameters reached effective sample size (ESS) > 1000.

## (h) Plastid phylogeny

A plastid dataset of 75 proteins [70] was extended by adding the 22 missing species to mimic the phylotranscriptomic nuclear tree (different species of the same genus were sometimes used). Homologous proteins were identified by BLASTP (*e*-value < $1 \times 10^6$) from available plastomes or transcriptomes. Genes were aligned with default MAFFT options and trees inferred with IQ-TREE under BIC-selected models and 1000 SH-aLRT. Alignments and gene trees were visualized with FigTree (https://github.com/rambaut/figtree) and SeaView [71] to remove paralogues and contaminants. Cleaned gene sequences were masked with PREQUAL, aligned with MAFFT (--allowshift --unalignlevel 0.8), and positions greater than 33% gaps removed with ClipKIT so that final alignments had lengths similar to Ruhfel *et al.* [70]. After concatenation, the plastid dataset consisted of 28 taxa and 16 085 aligned amino acid positions (32% missing data). Maximum-likelihood trees were inferred using IQ-TREE under BIC-selected homogeneous (cpREV + F + I + $\Gamma$4) and mixture (cpREV + F + I + $\Gamma$4 + C60) models and branch support assessed with 1000 UFBoot2 [62] and SH-like aLRT [63] pseudoreplicates.

## (i) Quantification of gene expression

Filtered and trimmed reads were mapped against the *Chlorokybus* genome (CCAC 0220) [13] using STAR v2.7.3a [25] (--runMode alignReads --outFilterMultimapNmax 20 --alignSJoverhangMin 8 --alignSJDBoverhangMin 1 --outFilterMismatchNmax 999 --outFilterMismatchNoverLmax 0.05 --alignIntronMin 20 --alignIntronMax 1 000 000 --alignMatesGapMax 1 000 000 --twopassMode Basic --sjdbScore 1 --quantMode TranscriptomeSAM --quantTranscriptomeBan IndelSoftclipSingleend). Gene expression was quantified with RSEM [26] (default parameters), followed by cross-sample normalization (TMM) using edgeR as implemented in Trinity (abundance_estimates_to_matrix.pl). Heatmaps were plotted ComplexHeatmap v2.6.2 in R v4.0.3 [72].

Data accessibility. RNAseq data are available on NCBI (Bioproject PRJNA708203). RNAseq FastQC reports, transcriptome assemblies, preliminary orthogroups, final concatenated alignments, phylogenetic trees, and molecular clock results are available from the Dryad Digital Repository: https://doi.org/10.5061/dryad.0gb5mkm25 [73].

Authors' contributions. I.I.: conceptualization, data curation, formal analysis, investigation, methodology, visualization, writing-original draft, writing-review and editing; T.D.: conceptualization, data curation, formal analysis, investigation, methodology, resources, visualization, writing-original draft, writing-review and editing; T.P.: formal analysis, investigation, methodology, resources, visualization, writing-review and editing; J.M.R.F.-J.: formal analysis, investigation; M.J.: formal analysis, writing-review and editing; J.d.V.: conceptualization, funding acquisition, project administration, resources, supervision, visualization, writing-original draft, writing-review and editing.

All authors gave final approval for publication and agreed to be held accountable for the work performed therein.

Competing interests. The authors declare no competing interests.

Funding. This work used the Scientific Compute Cluster at GWDG, the joint data centre of Max Planck Society for the Advancement of Science (MPG) and University of Göttingen. J.d.V. thanks the European Research Council for funding under the European Union's Horizon 2020 programme (Grant Agreement No. 852725; ERC-StG 'TerreStriAL') and the Deutsche Forschungsgemeinschaft (VR132/4-1). J.M.R.F-J. is supported by the 'Göttingen Graduate Center for Neurosciences, Biophysics, and Molecular Biosciences' (GGNB), University of Göttingen.

Acknowledgements. We thank Dr Maike Lorenz (SAG, Göttingen) for aid in culture obtainment, Prof. Dr Ute Krämer (RUB, Bochum) for access to the *Arabidopsis halleri* genome and Dr Michael Guiry (NUI, Galway) for help with nomenclatural questions. I.I., J.M.R.F-J., and J.d.V. are part of the MAdLand DFG priority programme 2237 (http://madland.science). We thank Debbie Maizels (www.scientific-art.com) for her superb figure 2.

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
