## [Peer Review File · Proceedings of the Royal Society B: Biological Sciences]

Review History

RSPB-2021-1425.R0 (Original submission)

Review form: Reviewer 1

Recommendation

Accept with minor revision (please list in comments)

Scientific importance: Is the manuscript an original and important contribution to its field?

Good

General interest: Is the paper of sufficient general interest?

Good

Quality of the paper: Is the overall quality of the paper suitable?

Good

Is the length of the paper justified?

Yes

Should the paper be seen by a specialist statistical reviewer?

No

Do you have any concerns about statistical analyses in this paper? If so, please specify them explicitly in your report.

No

It is a condition of publication that authors make their supporting data, code and materials available - either as supplementary material or hosted in an external repository. Please rate, if applicable, the supporting data on the following criteria.

Is it accessible?

Yes

Is it clear?

Yes

Is it adequate?

Yes

Do you have any ethical concerns with this paper?

No

Comments to the Author

This paper explores the diversity within the streptophyte lineage, Chlorokybophyceae, providing important omics data to identify at least five extant members. In my opinion, these analyses are incredibly important to understand the diversity within plant lineages as well as informing our understanding of the ancestral states of important traits during early plant evolution (e.g. stress responses, multicellularity, plant symbioses). Overall, this paper is concise and clearly written, providing a good overview of the complexities and nuances of understanding streptophyte evolution.

I have detailed a few comments, questions and clarifications below, going line by line.

In Figure 1, reference is made to "(A) *C. atmophyticus* ACOI 1086". However this doesn't seem to be present in the figure, with micrographs present for B-F. I presume they are just mislabelled in the figure legends as A-B and D-F.

In Line 71, reference is made to published genomic and transcriptomic information. However I think it would be good to make clear where this data has come from in a supplementary table. I may have missed it, but apart from Line 309, the specific repositories for the genome data are not mentioned in the text (e.g. for *Amborella trichopoda*, *Selaginella moellendorffii* etc). It would be good to see this mentioned in the text or a Supplementary Table.

In Line 72, I'm not entirely clear what 21% missing data means. Is this across the 487 loci a total of 21% of data is missing? If this is the case, does this vary across taxa particularly much?

To help the narrative, it might be useful to include, if possible, a summary of the divergence dates of Chlorokybophyceae in Figure 3, similar to Figure 1 from Del Cortona et al (2020) PNAS. Otherwise the divergence time analysis appears a bit out of nowhere.

I'm not sure if I missed something somewhere (quite possible), but the species sampling changes between Supplementary Fig 3 and Figure 3 which should be explained in the text. For example, *Gnetum montanum* is replaced by *Gnetum parvifolium*, whilst *Ostreococcus lucimarinus* and *Chara braunii* are removed.

In Figure 3b-c, it is not clear to me what TMM stands for in the context of gene expression differences. This should be explained in the text. Also in Figure 3b-c, it seems a little odd to me to

have the key in values of 50000 and 1e+05.

At Lines 168, 176, 189, 198 and 211, reference is made to Supplementary Figure 5. There doesn't appear to be a Supplementary Figure 5 in the Supplementary Information and I presume reference is being made to Supplementary Figure 1. This can be corrected in the text.

Finally Supplementary material 3 is not made reference to in the main text which should be corrected.

Review form: Reviewer 2

Recommendation

Major revision is needed (please make suggestions in comments)

Scientific importance: Is the manuscript an original and important contribution to its field?

Excellent

General interest: Is the paper of sufficient general interest?

Excellent

Quality of the paper: Is the overall quality of the paper suitable?

Excellent

Is the length of the paper justified?

Yes

Should the paper be seen by a specialist statistical reviewer?

No

Do you have any concerns about statistical analyses in this paper? If so, please specify them explicitly in your report.

No

It is a condition of publication that authors make their supporting data, code and materials available - either as supplementary material or hosted in an external repository. Please rate, if applicable, the supporting data on the following criteria.

Is it accessible?

Yes

Is it clear?

Yes

Is it adequate?

Yes

Do you have any ethical concerns with this paper?

No

Comments to the Author

This is a fantastic contribution to Proc. Roy. Soc. B. I very much enjoyed perusing your manuscript and some additional work will make this an outstanding paper. I am recommending major review mostly because my suggested changes will take you quite some time, even though

the quality of your contribution is undeniable. Please, find my comments in the pdf attached. Major concerns pertain the absence of Spirotaenia from all analyses and the nomenclatural treatment. This is a fairly complicated case (from a nomenclature perspective) and I urge you to reach out to a nomenclature specialist so that no further reviews are needed, and the case of Chlorokybophyceae is finally settled. Looking forward to your reviewed manuscript.

Decision letter (RSPB-2021-1425.R0)

11-Aug-2021

Dear Dr Irisarri:

I am writing to inform you that your manuscript RSPB-2021-1425 entitled "Unexpected cryptic species at the deepest branch of streptophytes" has, in its current form, been rejected for publication in Proceedings B.

This action has been taken on the advice of referees, who are both positive about the work but one of whom has recommended that substantial revisions are necessary. With this in mind we would be happy to consider a resubmission, provided the comments of the referees are fully addressed. However please note that this is not a provisional acceptance.

Sincerely,
Professor Hans Heesterbeek
mailto: proceedingsb@royalsociety.org

Reviewer(s)' Comments to Author:

Referee: 1

Comments to the Author(s)

This paper explores the diversity within the streptophyte lineage, Chlorokybophyceae, providing important omics data to identify at least five extant members. In my opinion, these analyses are

incredibly important to understand the diversity within plant lineages as well as informing our understanding of the ancestral states of important traits during early plant evolution (e.g. stress responses, multicellularity, plant symbioses). Overall, this paper is concise and clearly written, providing a good overview of the complexities and nuances of understanding streptophyte evolution.

I have detailed a few comments, questions and clarifications below, going line by line.

In Figure 1, reference is made to “(A) *C. atmophyticus* ACOI 1086”. However this doesn’t seem to be present in the figure, with micrographs present for B-F. I presume they are just mislabelled in the figure legends as A-B and D-F.

In Line 71, reference is made to published genomic and transcriptomic information. However I think it would be good to make clear where this data has come from in a supplementary table. I may have missed it, but apart from Line 309, the specific repositories for the genome data are not mentioned in the text (e.g. for *Amborella trichopoda*, *Selaginella moellendorffii* etc). It would be good to see this mentioned in the text or a Supplementary Table.

In Line 72, I’m not entirely clear what 21% missing data means. Is this across the 487 loci a total of 21% of data is missing? If this is the case, does this vary across taxa particularly much?

To help the narrative, it might be useful to include, if possible, a summary of the divergence dates of Chlorokybophyceae in Figure 3, similar to Figure 1 from Del Cortona et al (2020) PNAS. Otherwise the divergence time analysis appears a bit out of nowhere.

I’m not sure if I missed something somewhere (quite possible), but the species sampling changes between Supplementary Fig 3 and Figure 3 which should be explained in the text. For example, *Gnetum montanum* is replaced by *Gnetum parvifolium*, whilst *Ostreococcus lucimarinus* and *Chara braunii* are removed.

In Figure 3b-c, it is not clear to me what TMM stands for in the context of gene expression differences. This should be explained in the text. Also in Figure 3b-c, it seems a little odd to me to have the key in values of 50000 and 1e+05.

At Lines 168, 176, 189, 198 and 211, reference is made to Supplementary Figure 5. There doesn’t appear to be a Supplementary Figure 5 in the Supplementary Information and I presume reference is being made to Supplementary Figure 1. This can be corrected in the text.

Finally Supplementary material 3 is not made reference to in the main text which should be corrected.

Referee: 2

Comments to the Author(s)

This is a fantastic contribution to Proc. Roy. Soc. B. I very much enjoyed perusing your manuscript and some additional work will make this an outstanding paper. I am recommending major review mostly because my suggested changes will take you quite some time, even though the quality of your contribution is undeniable. Please, find my comments in the pdf attached. Major concerns pertain the absence of Spirotaenia from all analyses and the nomenclatural treatment. This is a fairly complicated case (from a nomenclature perspective) and I urge you to reach out to a nomenclature specialist so that no further reviews are needed, and the case of Chlorokybophyceae is finally settled. Looking forward to your reviewed manuscript.

Author's Response to Decision Letter for (RSPB-2021-1425.R0)

See Appendix A.

RSPB-2021-2168.R0

Review form: Reviewer 2

Recommendation

Accept with minor revision (please list in comments)

Scientific importance: Is the manuscript an original and important contribution to its field?

Excellent

General interest: Is the paper of sufficient general interest?

Excellent

Quality of the paper: Is the overall quality of the paper suitable?

Excellent

Is the length of the paper justified?

Yes

Should the paper be seen by a specialist statistical reviewer?

No

Do you have any concerns about statistical analyses in this paper? If so, please specify them explicitly in your report.

No

It is a condition of publication that authors make their supporting data, code and materials available - either as supplementary material or hosted in an external repository. Please rate, if applicable, the supporting data on the following criteria.

Is it accessible?

Yes

Is it clear?

Yes

Is it adequate?

Yes

Do you have any ethical concerns with this paper?

No

Comments to the Author

Dear authors,

First of all, I do realize I should have brought this up in my first round of review, my apologies. That said, the same way your former use of "ancestral" in the previous version of this fantastic manuscript could lead to misconstruing these fascinating organisms as ancestral (when they are not, no matter how many ancestral polymorphisms they may retain), if you write "at the deepest

branch" in your title (of all places), you are forgetting every node in a bifurcating tree is subtended by a single branch and that it results into two branches (hence, bifurcating). Meaning, as you well know (and I do realize I am preaching to the choir), the branch subtending the (Mesostigma, Spirotaenia, Chlorokybus) clade is as deep as the branch subtending the clade composed of the remainder of Streptophyta, which just happens to be more speciose. In other words, please, change the title. I'm thinking, maybe changing "branch" for "branching" would suffice. You decide but, please, do change the title.

Thank you.

Decision letter (RSPB-2021-2168.R0)

28-Oct-2021

Dear Dr Irisarri

I am pleased to inform you that your manuscript RSPB-2021-2168 entitled "Unexpected cryptic species at the deepest branch of streptophytes" has been accepted for publication in Proceedings B.

The referee and the Associate Editor have recommended publication, but also suggest some minor revision, in particular to the title of your manuscript. Therefore, I invite you to respond to the comment and revise your manuscript. Because the schedule for publication is very tight, it is a condition of publication that you submit the revised version of your manuscript within 7 days. If you do not think you will be able to meet this date please let us know.

Sincerely,

Professor Hans Heesterbeek

Associate Editor

Board Member

Comments to Author:

I think the paper has improved a lot, I'd like to congratulate the authors. I agree with the referee on the point about the title. I was wondering if the authors could suggest an alternative title which avoids terms like "deep", "basal", or similar. It is hard, I tried to think of a few, but I failed!

Reviewer(s)' Comments to Author:

Referee: 2

Comments to the Author(s).

Dear authors,

First of all, I do realize I should have brought this up in my first round of review, my apologies. That said, the same way your former use of "ancestral" in the previous version of this fantastic manuscript could lead to misconstruing these fascinating organisms as ancestral (when they are not, no matter how many ancestral polymorphisms they may retain), if you write "at the deepest branch" in your title (of all places), you are forgetting every node in a bifurcating tree is subtended by a single branch and that it results into two branches (hence, bifurcating). Meaning, as you well know (and I do realize I am preaching to the choir), the branch subtending the (Mesostigma, Spirotaenia, Chlorokybus) clade is as deep as the branch subtending the clade composed of the remainder of Streptophyta, which just happens to be more speciose. In other words, please, change the title. I'm thinking, maybe changing "branch" for "branching" would suffice. You decide but, please, do change the title.

Thank you.

Author's Response to Decision Letter for (RSPB-2021-2168.R0)

See Appendix B.

Decision letter (RSPB-2021-2168.R1)

01-Nov-2021

Dear Dr Irisarri

I am pleased to inform you that your manuscript entitled "Unexpected cryptic species among streptophyte algae most distant to land plants" has been accepted for publication in Proceedings B.

Your article has been estimated as being 9 pages long. Our Production Office will be able to confirm the exact length at proof stage.

Data Accessibility section

Open Access

Paper charges

Sincerely,

Proceedings B

Appendix A

Referee: 1

Comments to the Author(s)

This paper explores the diversity within the streptophyte lineage, Chlorokybophyceae, providing important omics data to identify at least five extant members. In my opinion, these analyses are incredibly important to understand the diversity within plant lineages as well as informing our understanding of the ancestral states of important traits during early plant evolution (e.g. stress responses, multicellularity, plant symbioses). Overall, this paper is concise and clearly written, providing a good overview of the complexities and nuances of understanding streptophyte evolution.

>We thank the reviewer for all the thoughtful comments, which we hope to have appropriately addressed.

I have detailed a few comments, questions and clarifications below, going line by line.

In Figure 1, reference is made to “(A) *C. atmophyticus* ACOI 1086”. However this doesn’t seem to be present in the figure, with micrographs present for B-F. I presume they are just mislabelled in the figure legends as A-B and D-F.

>Corrected.

In Line 71, reference is made to published genomic and transcriptomic information. However I think it would be good to make clear where this data has come from in a supplementary table. I may have missed it, but apart from Line 309, the specific repositories for the genome data are not mentioned in the text (e.g. for *Amborella trichopoda*, *Selaginella moellendorffii* etc). It would be good to see this mentioned in the text or a Supplementary Table.

>The source of genomic data is now available in the new supplementary Table 3, referenced in the Materials and Methods.

In Line 72, I’m not entirely clear what 21% missing data means. Is this across the 487 loci a total of 21% of data is missing? If this is the case, does this vary across taxa particularly much?

>The new dataset has 17% missing data, calculated as the proportion of empty cells in a taxon vs. loci matrix. The proportion of missing data per species is also detailed in Supplementary Table 3. With the exception of *Chaetosphaeridium*, all other taxa had $\geq 70\%$ loci present.

To help the narrative, it might be useful to include, if possible, a summary of the divergence dates of Chlorokybophyceae in Figure 3, similar to Figure 1 from Del Cortona et al (2020) PNAS. Otherwise the divergence time analysis appears a bit out of nowhere.

>We agree that the divergence time analyses might have come across a bit out of context. We now provide a detailed explanation of the most important divergence times of Chlorokybus and Arabidopsis in the main text and comment on the effect of applying alternative calibration methods (as per request of Referee 2). Further details are provided in the new Supplementary Fig. 4.

I'm not sure if I missed something somewhere (quite possible), but the species sampling changes between Supplementary Fig 3 and Figure 3 which should be explained in the text. For example, *Gnetum montanum* is replaced by *Gnetum parvifolium*, whilst *Ostreococcus lucimarinus* and *Chara braunii* are removed.

>The Referee is right. The sampling of species changed slightly between the nuclear and plastid dataset simply due to the availability of plastomes. This is now clarified in the Materials and Methods: A plastid dataset of 75 proteins was extended by adding the 22 missing species to mimic the phylotranscriptomic nuclear tree (different species of the same genus were sometimes used)"

In Figure 3b-c, it is not clear to me what TMM stands for in the context of gene expression differences. This should be explained in the text. Also in Figure 3b-c, it seems a little odd to me to have the key in values of 50000 and 1e+05.

>TMM stands for "trimmed mean of M-values" and unlike FPKM or RPKM, it is a between-sample normalization method. This is now explained in the main text. We have also homogenized the scale in Fig. 3B-C.

At Lines 168, 176, 189, 198 and 211, reference is made to Supplementary Figure 5. There doesn't appear to be a Supplementary Figure 5 in the Supplementary Information and I presume reference is being made to Supplementary Figure 1. This can be corrected in the text.

>Thank you for spotting this error, which we amended.

Finally Supplementary material 3 is not made reference to in the main text which should be corrected.

>Thank you for spotting this too. We moved this section to the main text as Box 1.

Referee: 2

Comments to the Author(s)

This is a fantastic contribution to Proc. Roy. Soc. B. I very much enjoyed perusing your manuscript and some additional work will make this an outstanding paper. I am recommending major review mostly because my suggested changes will take you quite some time, even though the quality of your contribution is undeniable. Please, find my comments in the pdf attached.

>Thank you for all the thoughtful and careful comments that have helped us improve the manuscript.

Major concerns pertain the absence of Spirotaenia from all analyses and the nomenclatural treatment. This is a fairly complicated case (from a nomenclature perspective) and I urge you to reach out to a nomenclature specialist so that no further reviews are needed, and the case of Chlorokybophyceae is finally settled. Looking forward to your reviewed manuscript.

>Following the Reviewer's suggestion, we now added Spirotaenia minuta in our plastid and nuclear analyses, which provides an important reference for Chlorokybus. Given that the taxonomic status of Spirotaenia remains unsettled, we focused the taxonomic aspects on our focal group Chlorokybophyceae. We also carefully checked with expert colleague the nomenclatural aspects for the descriptions of new species and the higher taxonomic ranks.

Abstract. You seem to be implying these taxa are "ancestral", which I'm hoping it is not the case. Please, see the following and remove ambiguity. Thank you.

>Thank you for the important remark. We changed the sentence to "The apparent low diversity in these key lineages complicates macroevolutionary studies that aim to reconstruct the early evolution of key traits in the streptophyte ancestor".

Systematic botany. You need to reach out to a nomenclature specialist. There are far too many concerning issues with this section.

>Thank you for the detailed comments on the nomenclature, which we have now discussed with an expert colleague and amended in the paper.

Firstly, explicitly state author's for all new names. The paper's authors need not be the species authors. It is vital to remove all semblance of ambiguity. Also, if you want to keep the name, give credit to who first proposed it, that is, use "ex". The revised class name would be "Chlorokybophyceae K.Bremer ex YOU_GO_HERE"

>The authorship of the species are those of the manuscript, thus there is no name of specifying them for each of the species.

Points 9 and 10 of article 10 of the Shenzhen ICN (here https://www.iapt-taxon.org/nomen/pages/main/art_10.html) state (Turland et al. 2018):

10.9. The type of a name of a family or of any subdivision of a family is the same as that of the generic name from which it is formed (see Art. 18.1). For purposes of designation or citation of a type, the generic name alone suffices. The type of a name of a family or subfamily not formed from a generic name is the same as that of the corresponding alternative name (Art. 18.5 and 19.8).

10.10. The principle of typification does not apply to names of taxa above the rank of family, except for names that are automatically typified by being formed from generic names (see Art. 16.1(a)), the type of which is the same as that of the generic name.

Additionally, I have doubts about the validity of Chlorokybaceae and Chlorokybales. Neither these two nor Chlorokybophyceae were accompanied by either a formal description or "by a reference to a previously and effectively published description or diagnosis". That is, a page number is needed and none is provided neither in Rogers et al. (1980) nor in Bremer (1985). Meaning, order Chlorokybales and family Chlorokybaceae, proposed by Rogers et al. (1980), would be invalid for the same reasons that apply to Chlorokybophyceae.

From Rogers et al. (1980): "We propose that Chlorokybus be classified in the - Chlorokybaceae and Chlorokybales of the Charophyceae sensu lato (Stewart and Mattox, 1975). Since Chlorokybus is the only genus to be presently included in the order and family, we formally base the family on the Latin de- scription of Chlorokybus by Geitler (1942)." [notice that, although a reference is given, the page number is missing and there is NO description]

From Bremer (1985): "Subdivision Chlorokybophytina based on Class Chlomkybophyceae based on Chlorokybales Mattox and Stewart (1984), Chlorokybus". [notice that, although a reference is given, the page number is missing and there is NO description]

See articles 38.1 and 41.5 of the International Code of Nomenclature (ICN) for algae, fungi, and plants (Turland et al. 2018).

Article 38.1 of the ICN (https://www.iapt-taxon.org/nomen/pages/main/art_38.html#Art38.1) states: In order to be validly published, a name of a new taxon (see Art. 6.9) must (a) be accompanied by a description or diagnosis of the taxon (see also Art. 38.7 and 38.8) or, if none is provided in the protologue, by a reference (see Art. 38.13) to a previously and effectively published description or diagnosis (except as provided in Art. 13.4 and H.9; see also Art. 14.9 and 14.14); and (b) comply with the relevant provisions of Art. 32–45 and F.4–F.5."

Article 41.5 of the Shenzhen ICN (https://www.iapt-taxon.org/nomen/pages/main/art_41.html#Art41.5) states: On or after 1 January 1953, a new combination, name at new rank, or replacement name is not validly published unless its basionym or replaced synonym is clearly indicated and a full and direct reference given to its author and place of valid publication, with page or plate reference and date (but see Art. 41.6 and 41.8). On or after 1 January 2007, a new combination, name at new rank, or replacement name is not validly published unless its basionym or replaced synonym is cited.

>We thank the reviewer for this detailed account. Following the reviewer's suggestions, we have re-described the class Chlorokybophyceae, order Chlorokybales, and Family Chlorokybophyceae.

Please, provide etymology for all new names

>Etymology statements have been added.

Please, deposit duplicates elsewhere

>We submitted our new strains to CCAP and will add the vouchers.

Methods. How did you plot your phylogenies? What software did you rely on? Please, cite software used so that developers can justify funding needs in their grant applications. If you relied on R, cite the project version you used, as well as packages, with their respective references.

>Thank you; we have checked appropriate citation to all software (FigTree, R, etc.).

Please, be aware the NovaSeq platform relies on 2-channel chemistry for sequencing, which can result in "an overrepresentation of poly-G-mers towards the end of the reads" (see De-Kayne et al. 2020, <https://doi.org/10.1111/1755-0998.13309>) regrettably, trimmomatic does not account for potential poly-G-mer artifacts derived from 2-channel sequencing chemistry (fastp is a good alternative <https://github.com/OpenGene/fastp>)

>The Referee is very right and we observed the overrepresentation of Gs at 3' in other NovaSeq runs. However, this was not the case for the current runs. The FastQC reports of our reads are now available in FigShare.

You could mine 1KP's *Spirotaenia minuta* (<https://trace.ncbi.nlm.nih.gov/Traces/sra/?run=ERR364381>) for SSU and ITS (e.g., using HybPiper, <https://github.com/mossmatters/HybPiper/wiki>)
How are you rooting this tree? *Spirotaenia* would come in very handy in here...

>We included *Spirotaenia* in our phylotranscriptomic nuclear and plastid phylogenies, but not in the SSU phylogeny, where the genetic distance to *Chlorokybus* spp. was extremely large (>15% vs. 1% within *Chlorokybus*). This produced random rooting effects and inconsistent results under various phylogenetic inference methods and alignment trimming algorithms. *Spirotaenia*'s ITS2 region was particularly divergent, but the removal of ITS2 did not solve the problem, perhaps because too few variable sites remained within *Chlorokybus*. Therefore, we rooted the SSU phylogeny using the root obtained from the larger phylotranscriptomic dataset (as stated now in Fig 1's caption).

Please, implement a MSC approach for species tree building such as ASTRAL, in addition to your total evidence approach, which does not account for ILS and an infinity of other known biological processes. Additionally, methods like ASTRAL are more robust to LBA artifacts and, even, the presence of paralogs (Yan et al. 2021, <https://doi.org/10.1093/sysbio/syab056>)

>We now implement a MSC analysis with ASTRAL.

Remove “Deep coalescence”. Please, see Doyle (2021). *Syst. Biol.*
(<https://doi.org/10.1093/sysbio/syab053>)

>We thank the reviewer for noticing this important point, which we have corrected.

Why not just mask third-codon position across your alignments?

>Removing third-codon positions is an effective measure to reduce sequence saturation when inferring deep divergences, but so is the use of amino acid data, which are mostly determined by the first two codon positions anyway. In addition, using amino acids provides a larger set of possible states (20 vs 4), facilitating multiple sequence alignments and allowing the use of complex mixture models that are robust to phylogenomic artifacts by accounting for site-specific frequencies.

Molecular clock. Why not gamma priors? See Foster et al. 2017
(<https://doi.org/10.1093/sysbio/syw086>) and Nie et al. 2019
(<https://doi.org/10.1093/sysbio/syz032>)

>Following the Reviewer’s suggestion, we explored the effect of using more informative prior distributions for the eight fossil calibrations. Following Morris et al. 2018, we implemented truncated-cauchy and skew-t distributions, which compared to uniform distribution put more weight closer to the fossil age. The results are shown in Supplementary Fig. 4. As mentioned in the main text, the use of more informative priors produced slightly younger divergences (as expected) but differences were not substantial, particularly within Chlorokybus. Such sensitivity analyses were important to understand the effect of methodological choices, but in this case we agree with Morris et al. 2018: available information on plant and algae fossil is insufficient to favour more informative priors over the more conservative uniform distributions.

Could you, please, elaborate? Also, just in case this paper is not in your radar:

– Mongiardino Koch (2021). Phylogenomic Subsampling and the Search for Phylogenetically Reliable Loci. *MBE* (<https://doi.org/10.1093/molbev/msab151>)

Basically, you could select a subset of representative genes (gene shopping approach) and run regular old BEAST with partitions and everything.

>We used the entire phylotranscriptomic dataset to obtain rate parameters according to a single LG+G model, which we favoured to obtain precise model parameters. Multiple alternatives are possible, including using multiple data partitions in MCMCTREE or selecting a subset of clock-like genes to be able to run other software like BEAST or PhyloBayes. We have explored all these alternatives in previous studies (e.g., Strassert et al. 2021 *Nat Commun*: <https://www.nature.com/articles/s41467-021-22044-z>) and found that MCMCTREE with a single data partition produces effective and reliable time estimates. Our goal here is to provide an approximate estimation of the absolute divergences within Chlorokybus, which remains consistent across the explored analyses. A comparison of methodological choices in inferring divergence times across the green lineage is undeniably interesting, but beyond scope.

Appendix B

Response to Referees

Associate Editor

I think the paper has improved a lot, I'd like to congratulate the authors. I agree with the referee on the point about the title. I was wondering if the authors could suggest an alternative title which avoids terms like "deep", "basal", or similar. It is hard, I tried to think of a few, but I failed!

>We are pleased to read that all previous concerns were properly addressed. We agree that the original title could be misunderstood, and therefore we have changed it to: "Unexpected cryptic species among streptophyte algae most distant to land plants".

Referee: 2

Dear authors,

First of all, I do realize I should have brought this up in my first round of review, my apologies. That said, the same way your former use of "ancestral" in the previous version of this fantastic manuscript could lead to misconstruing these fascinating organisms as ancestral (when they are not, no matter how many ancestral polymorphisms they may retain), if you write "at the deepest branch" in your title (of all places), you are forgetting every node in a bifurcating tree is subtended by a single branch and that it results into two branches (hence, bifurcating). Meaning, as you well know (and I do realize I am preaching to the choir), the branch subtending the (Mesostigma, (Spirotaenia, Chlorokybus) clade is as deep as the branch subtending the clade composed of the remainder of Streptophyta, which just happens to be more speciose. In other words, please, change the title. I'm thinking, maybe changing "branch" for "branching" would suffice. You decide but, please, do change the title. Thank you.

>Thank you for bringing up this point about the original title. We fully agree with the criticism and we hope to have corrected it.